# Silica Nanoparticles in Xanthan Gum Solutions: Oil Recovery Efficiency in Core Flooding Tests

**DOI:** 10.3390/nano13050925

**Published:** 2023-03-02

**Authors:** Dayan L. Buitrago-Rincon, Véronique Sadtler, Ronald A. Mercado, Thibault Roques-Carmes, Philippe Marchal, Samuel F. Muñoz-Navarro, María Sandoval, Julio A. Pedraza-Avella, Cécile Lemaitre

**Affiliations:** 1Grupo de Investigación en Fenómenos Interfaciales, Reología y Simulación de Transporte (FIRST), Universidad Industrial de Santander, Bucaramanga 680002, Colombia; 2Laboratoire Réactions et Génie des Procédés, Université de Lorraine, CNRS, F-54000 Nancy, France; 3Grupo de Investigación en Recobro Mejorado (GRM), Universidad Industrial de Santander, Bucaramanga 680002, Colombia

**Keywords:** xanthan gum, silica nanoparticles, HPAM, core flooding, rheology, %EOR

## Abstract

Polymer flooding is one of the enhanced oil recovery (EOR) methods that increase the macroscopic efficiency of the flooding process and enhanced crude oil recovery. In this study, the effect of silica nanoparticles (NP-SiO_2_) in xanthan gum (XG) solutions was investigated through the analysis of efficiency in core flooding tests. First, the viscosity profiles of two polymer solutions, XG biopolymer and synthetic hydrolyzed polyacrylamide (HPAM) polymer, were characterized individually through rheological measurements, with and without salt (NaCl). Both polymer solutions were found suitable for oil recovery at limited temperatures and salinities. Then, nanofluids composed of XG and dispersed NP-SiO_2_ were studied through rheological tests. The addition of nanoparticles was shown to produce a slight effect on the viscosity of the fluids, which was more remarkable over time. Interfacial tension tests were measured in water-mineral oil systems, without finding an effect on the interfacial properties with the addition of polymer or nanoparticles in the aqueous phase. Finally, three core flooding experiments were conducted using sandstone core plugs and mineral oil. The polymers solutions (XG and HPAM) with 3% NaCl recovered 6.6% and 7.5% of the residual oil from the core, respectively. In contrast, the nanofluid formulation recovered about 13% of the residual oil, which was almost double that of the original XG solution. The nanofluid was therefore more effective at boosting oil recovery in the sandstone core.

## 1. Introduction

Enhanced oil recovery (EOR) processes, including polymer flooding, are employed to significantly improve the oil recovery factor [1,2]. These chemical methods consist of the injection of a displacing fluid in oil reservoirs to mobilize the crude oil trapped in the porous rocks. Typically, the displacing fluid contains a water-soluble polymer (polymer flooding) [3,4]. The mechanism involved in polymer flooding is based on decreasing the water/oil mobility ratio (M): the mobility difference between displacing (polymer flooding) and displaced fluids (oil). The water/oil mobility ratio is inversely proportional to the viscosity, so the higher the viscosity of the fluid, the lower the value of M. The displacement of the oil by fluid will occur in a piston-like flow, but with low viscosity, the waterflooding will finger through the oil causing formation damage and plugging [5,6,7].

The main criteria to choose EOR polymers are: very high molecular weight, inexpensive, thickener capacity, and complete solubility in water. Additionally, aqueous solutions should be able to tolerate high salinity, high temperatures, and resistance to viscosity degradation over time [8]. The rheological properties of a polymer solution are strongly dependent on the polymer structure, composition, conformation, and interactions between the polymer and the solvent. The most classic polymeric system employed for EOR is hydrolyzed polyacrylamide (HPAM), followed by xanthan gum (XG). XG polymer is a BioSource alternative to conventional flooding methods. However, XG alone has shown limitations in these processes due to its degradation over time, accelerated by temperature [2,8,9].

HPAM is a co-polymer of acrylamide and acrylic acid, Figure 1a. This polyelectrolyte is exhibiting negative charges on its chains. HPAM solutions significantly improve the microscopic oil displacement efficiency, thanks to their synthetic nature and adaptability [10,11,12]. The molecular chains of HPAM adjust easily during flow under shear rate. Indeed, the backbone of HPAM is flexible. However, the HPAM carboxyl groups are extremely sensitive to salts, especially high valence salts. Inorganic salts can compress the double electrode layer of this polyelectrolyte, weakening the static repulsion between the HPAM lateral groups. The conformational transition becomes more constricted, and the viscosity of the polymer solution decreases sharply. The chemical modification of HPAM through the addition of specific lateral groups decreases the sensitivity of the molecules to salt [2,13,14].

XG is an anionic heteropolysaccharide produced by the microorganism *xanthomonas campestris*. XG is a relatively cheap biopolymer and is suitable for oil recovery at limited temperatures and salinities [14]. It is mainly composed of a linear β-D linked glucose backbone with a trisaccharide side chain on every other glucose at C-3. In aqueous solutions, xanthan macromolecules can exhibit a disordered conformational structure that tends to maximize the solution viscosity, Figure 1b [2,15]. In its ordered conformation, it exhibits a rigid structure and a reduction in fluid viscosity. XG molecules may become rigid when an amount of electrolyte is present, which leads the side chains to collapse around the backbone. Nonetheless, the XG biopolymers have a wide range of applications in many industries due to their thickener properties, high salinity tolerance (more than HPAM), thermal stability, affordability, safety (non-toxic), as well as its low environmental impact.

This leads to the motivation for this study. Green polymers such as XG have the potential for EOR processes. However, its main limitation has been its biodegradability in aqueous solutions under the adverse conditions of pressure, temperature, and ionic strength occurring in the field. To improve the efficiency of XG solutions, materials such as silica nanoparticles (NP-SiO_2_) are added, improving the rheological behavior and maintaining its stability over time at high salinity and high-temperature conditions due to weak physical attractive forces (hydrogen bonds) between XG and NP-SiO_2_. The XG/NP-SiO_2_ interactions change the conformation of the polymer, which can result in improved microscopic and macroscopic sweep efficiencies [16,17,18].

In a previous study [17], different methods were investigated for the preparation of nanofluids composed of hydrocolloid XG and hydrophilic silica nanoparticles (NP-SiO_2_) dispersed in water and brine. It was determined that the order of the addition of the components, the hydration time of the polymer XG, and the interaction time of the XG with the nanoparticles directly affected the resulting viscosity profile of the fluid. The method producing the highest viscosity at a shear rate of 7.3 s^−1^ was selected. The presence of nanoparticles caused an increase of 30% of up to XG solution viscosity. In addition, the polymeric dilution curve of XG at high salinity concentrations, 3% NaCl, was determined. This curve allowed two differentiated working zones for EOR processes: the diluted zone (up to 600 ppm XG) and the semi-diluted zone (from 600 ppm to 4800 ppm XG). It was determined that the effect of the nanoparticles was more noticeable (on the rheological response) in the semi-dilute zone: higher viscosity and better stability were achieved in the presence of nanoparticles.

In a second previous study, the effect of silica nanoparticles (NP-SiO_2_) in XG solutions was investigated through the analysis of viscosity profiles and their stability over time. The nanofluids composed of XG and NP-SiO_2_ dispersed in water and brine were studied through two different aging tests (1 month and 7 months), at a constant storage temperature (60 °C). The addition of nanoparticles was shown to produce a slight increase in the viscosity of the fresh fluids (initial time = 0), while a more remarkable effect was observed over time: the viscosity diminution in time is significantly limited by the NP-SiO_2_. The presence of NP-SiO_2_ stabilizes the polymer solution, maintaining its viscosity level over time, due to a delay in the movement of the molecule. This indicates delayed biodegradation of the molecule at high temperatures and high ionic strengths in the solution. In this previous study, the molecular interaction between the XG molecule and the NP-SiO_2_ surface was demonstrated. The presence of NP-SiO_2_ increases the hydrodynamic radius of the polymer, which indicates attractive forces between these two components, maintaining colloidal stability with average size aggregates lower than 300 nm.

In the current paper, core flooding experiments are presented to further investigate the potential of nanofluids (XG + NP-SiO_2_) for enhanced oil recovery. First, the viscosity of XG polymeric solutions with and without nanoparticles was evaluated and compared to that of HPAM polymeric solutions (the most used polymer in EOR processes). Then, the effect of nanoparticles on interfacial tension (brine/polymer/mineral oil) was determined. Finally, the displacement efficiency of the nanofluids at reservoir conditions was evaluated and compared to the performance of XG and HPAM solutions without nanoparticles. This study aims to contribute to filling the knowledge gap in the nanofluid literature for EOR applications.

## 2. Materials and Methods

### 2.1. Materials

The xanthan gum was purchased from Sukin Industries. A commercial HPAM polymer, FLOPAAM 3230 S (hydrolysis degree: 30% at 25 °C), was obtained from SNF Floerger. The silica nanoparticles were fumed Aerosil 300 amorphous hydrophilic (300 ± 30 m^2^/g) from Evonik Industries. Sodium chloride was supplied by Sigma-Aldrich 99.5% (58.44 g/moL). Deionized water was used as a solvent. USP mineral oil was supplied by Elementos químicos LTDA (0.867 g/cm^3^ @ 25 °C and 8.46 cP @ 60 °C).

### 2.2. Fluids Preparation

All formulations were prepared according to the same method. In a previous study [17], the preparation method was selected and optimized for both polymeric solutions and nanofluids. Polymer solutions were prepared by incorporating polymer powder (XG or HPAM) in deionized water or 3% NaCl brine (previously prepared). A stirring time of 24 h was determined to ensure complete hydration of the polymers. For the nanofluids preparation, the polymer and the nanoparticles were introduced at the same time in brine (previously prepared). Then, the resulting fluid was mixed for 24 h (with a magnetic stirrer) to ensure complete polymer hydration and particle-polymer interaction. This method was the one yielding the best results, reaching the highest viscosity profile, with a complete interaction between the XG molecule and the silica nanoparticles [17]. In another previous study [18], the rheological behavior and the stability of the fluids were evaluated. The best results were obtained for an XG concentration of 1000 ppm (semi-dilute regime), and a fixed concentration of 300 ppm for nanoparticles.

### 2.3. Viscosity Behavior

The viscosity behavior of the studied polymer solutions was established using Anton Paar 302 rheometer with cone/plate geometry. The tests were executed at a constant temperature (60 °C) and for variable shear rate (0.1–100 [s^−1^]), with sensitivities of 1 Pa·s for the viscosity.

### 2.4. Interfacial Tension

The effect of the NP-SiO_2_ on the interfacial tension (IFT) between mineral oil and polymer solutions was evaluated. The Du Noüy ring method was used at a constant temperature (60 °C). The “weighting from below” practice was used in all the experiments since it proved to yield the most-probable reproducibility, equal to 0.002 mN/m. Three sets of measurements were conducted to determine oil-fluid interfacial tension alterations caused by nanoparticles: brine/mineral oil, XG solution/mineral oil, and nanofluid (XG + NP-SiO_2_)/mineral oil. The fluids were previously heated up to 60 °C and maintained at this condition during the tests.

### 2.5. Core Flooding Experiments

Three core flooding experiments were conducted to evaluate the displacement efficiency of the XG solutions, HPAM solutions and nanofluids (XG + NP-SiO_2_). They were performed on the STEAM LDE equipment [19]. Initially, petrophysical properties, such as effective porosity and absolute permeability, were determined by deionized water injection using mass balance and the Darcy equation. Subsequently, each porous medium was saturated with brine followed by a mineral oil injection process until irreducible water conditions were reached. Finally, the fluids being evaluated were injected into each porous medium and the recovery factor was calculated using mass balances. These three stages are described as core preparation, brine flooding, and fluid flooding. Figure 2 describes the experimental setup.

#### 2.5.1. Core Preparation

For the construction of the porous media, four core plugs of 3.86 cm internal diameter and 36.5 cm length were selected. The core plugs were packed with Ottawa sand using the same grain aggregation and ordering process, ensuring the reproducibility of the porous medium. The sand used was mainly composed of quartz with a density of 2.65 g/cm^3^. After its construction, each core plug was assembled and confined within the core holder and mounted on the STEAM LDE equipment for the execution of the subsequent displacement tests. Once the core was installed, the outer surface was pressurized to simulate the same pressure as in the oil reservoir.

The volume and theoretical porosity of the porous medium was calculated from the mass of sand in each packing and the density of quartz. The effective porosity of the core was determined by means of the mass balance (eliminating dead volumes) of the injection of 5 porous volumes (PV) of deionized water into the porous medium. The absolute permeability was calculated with Darcy’s law, which models the flow of fluids through a porous medium.
(1)$$Q=\frac{K\Delta PA_{T}}{245µL}$$
where *Q* is the flow rate (cm^3^/min), *K* the permeability (mD), Δ*P* the pressure loss (psi), *A_T_* the flow cross-sectional area (cm^2^), µ the fluid viscosity (cP), *L* the length of the medium (cm) and a conversion factor of 245. Equation (1) is reorganized into
(2)$$\frac{245\;Q}{A_{T}}=K\frac{\Delta P}{µL}$$

The absolute permeability of the core *(K)* was determined by measuring the slope of 245 *Q/A_T_* as a function of Δ*P*/(*µ**L*) [19,20].

#### 2.5.2. Reservoir Saturation Conditions

In this study, each core is prepared under particular reservoir conditions: temperature of 60 °C, formation brine 3% NaCl (30,000 ppm), and mineral oil.

In the brine saturation stage, a 3% NaCl solution (prepared with deionized water) was applied to saturate each core (3 PV), and, thereafter, the cores were flooded with mineral oil until obtaining an irreducible water saturation (Swirr) of the cores and a maximum oil saturation in the core. When Swirr conditions were reached, the effective permeability of the core to mineral oil was obtained with the measured pressure difference. Subsequently, the relative permeability (*K_ro_*) of the core under these conditions results from the ratio between the effective permeability of mineral oil (*K_eff_*) and the absolute permeability (*K_abs_*), [19].
(3)$$K_{ro\;@\;Swirr\;}=\frac{K_{eff}}{K_{abs}}$$

Finally, the mineral oil saturated cores were aged at 1050 psia pore pressure, for several hours.

#### 2.5.3. Flooding Process

In the final stage, the interaction of each fluid with the porous medium was estimated. Each fluid was evaluated successively in a different core under the same conditions: brine in Core 1, XG solution in Core 2, HPAM solution in Core 3, and nanofluid in Core 4. One PV of 3% NaCl brine was injected at a rate of 0.2 cm^3^/min, then each core was flooded with its own fluid. An injection rate of 0.2 cm^3^/min, 60 °C temperature and 1050 psi pressure were applied. The displacement efficiency was calculated by dividing the displaced oil volume by the initial oil volume.

## 3. Results and Discussion

### 3.1. Viscosity Behavior

The viscosity curves of XG solutions and HPAM solutions with and without NaCl (30,000 ppm) are compared to that of nanofluid (XG + NP-SiO_2_) in brine.

All the viscosity curves obtained show a marked shear-thinning behavior at 1000 ppm polymer concentration, Figure 3. The viscosities of the polymer solutions in deionized water are higher than those of solutions in brine. This is due to the fact that polymer chains are stretched in deionized water due to the repulsive forces between the negative charges (carboxylate groups) on the chain [21]. A stronger viscosity decrease is observed in brine for HPAM formulations. At a representative shear rate of 7.3 s^−1^, the viscosity of XG solutions decreases by 32.5% while the viscosity of HPAM solutions decreases by 50%. A possible explanation is the solvation and modification of the intermolecular hydrogen bonds of the polymers in saline solution. These macromolecular interactions change the polymer conformation, which can result in a viscosity reduction. For HPAM, the Na^+^ concentration in the aqueous solution prevents static repulsion effects on polyelectrolyte lateral groups, leading to electrostatic shielding and to a more contracted conformation. The hydrodynamic size of the polymer decreases, and the fluid viscosity is lower. On the other hand, in a saline medium, the transitional conformation of the XG changes into a more rigid structure. The electrostatic shielding effect of Na^+^ causes the lateral groups on the main chains to twine around the backbone and, due to this behavior, the conformation of XG converts into a rigid rod. The hydrophilic groups on the backbone are shielded and reduce the hydrodynamic size of the molecule, leading to a viscosity loss [8,22,23].

The same phenomenon was observed in previous works on the influence of NaCl on the preparation and performance of nanofluids. The presence of Na^+^ and Cl^−^ ions limits electrostatic repulsions between charges within the polymer XG, even in the presence of the nanoparticles, promoting a more rigid and less stretched backbone of the polymer/nanoparticle, resulting in a decrease in viscosity of the nanofluid [18,24,25,26,27,28].

### 3.2. Interfacial Tension

A correct knowledge of interfacial tension (IFT) and the viscosity of fluids used in EOR are necessary to efficiently manage the production process of field [29].

The mechanism of EOR involved in polymer/nanofluids flooding is based on decreasing the mobility difference between displacing (polymer solutions) and displaced fluids (petroleum) in order to reduce lingering effects. When working with improved polymeric fluids, in this case XG, with the incorporation of silica nanoparticles, it is important to evaluate the effect of this system on the tension properties at the interface with oil prototypes. The polymer should act as an effective viscosifier for the aqueous phase, and its effect in IFT is usually less pronounced than for surfactant fluids [30,31].

In this study, little effect was found on the interfacial properties when adding XG polymer in brine with or without silica nanoparticles. Table 1 shows very high interfacial tension in all the fluids tested at these conditions. This lack of surface activity is usually explained by a very slow equilibration of polymeric micelles, which prevents the possibility of the macromolecules migrating to the interfaces and affecting the IFT. In conclusion, it could be difficult for both the XG solution and the nanofluid (XG + NP-SiO_2_) to create stable emulsions with the oil under flow conditions [31].

### 3.3. Core Flooding Results

Core flooding simulates reservoir conditions and, therefore, contributes to investigating the EOR performances of polymer systems. Table 2 shows the obtained experimental data. The permeability was found to be similar in all the cores, between 2.3 and 2.7 D. Under waterflooding (3% NaCl brine), chosen as the reference case, 63.61% of the Initial Oil In Place (IOIP) was recovered. The displacement efficiency obtained under the flood of XG solution, HPAM solution, and nanofluid was, respectively, 70.27%, 76.79%, and 71.11% of IOIP, Figure 4. These data evidence that the addition of polymer molecules, such as HPAM and XG, in the initial stage allows for reaching larger oil recovery efficiencies compared to simple waterflooding. However, the oil recovery factor through nanofluid injection surpasses by far that obtained with the polymer solutions. The incremental oil recovery (laboratory conditions) for the nanofluid was 13.18%, and only 7.5% and 6.6% for HPAM and XG solutions, respectively.

Contrasting the results of the viscous behavior of the fluids (Figure 2 and Figure 3) and the displacement efficiency achieved by them in the core (Table 2), the direct relationship between the viscosity and the mobility ratio in the porous medium is verified: the mobility ratio is inversely proportional to the viscosity of the flooding solution.

The viscosity profiles of nanofluids were higher than those of polymeric solutions (Figure 3), therefore indicating higher displacement efficiency. In fact, the laboratory incremental percentages achieved by the nanofluid were almost double those obtained by the two evaluated polymeric solutions (Table 2).

In a previous study [18], Figure 5 was obtained. It was found that the effect of silica nanoparticles on XG solutions promotes fluid stability, by maintaining the viscosity profile: XG polymer solutions without nanoparticles and XG solutions with nanoparticles (nanofluids) were prepared and aged for this test. The fluids were stored for 7 months aging time at a constant temperature (60 °C), representative of the reservoir conditions. After 7 months, all the polymer solutions showed a strong decrease in viscosity, while the nanofluids displayed a moderate viscosity loss. The XG polymer solution without nanoparticles underwent a 72% viscosity reduction, while the same formulation containing nanoparticles exhibited a viscosity drop of only 34%. These results showed the potential of nanofluids (XG + NP-SiO_2_) for EOR processes [18].

For the present study, aged fluids were not evaluated. All formulations (polymeric solutions and nanofluids) were fresh (aging time = 0). Under these conditions, the HPAM solution was found to performed better than the XG solution. This could be due to the fact that HPAM solutions are more elastic than XG solutions at low concentrations. Thus, in porous media at this concentration, the higher elasticity of HPAM promotes a little higher displacement efficiency than XG polymer. [32,33,34]. But the nanofluids were found to perform way better than HPAM and XG solutions. Indeed, both the displacement efficiency and the laboratory incremental EOR for the nanofluid were higher than for the polymeric solutions. The incremental EOR performed by the nanofluids was 99.7% higher than that of XG solutions, and 75.7% higher than that of HPAM solution.

One possible explanation for the performance of the nanofluids is the effect of silica nanoparticles on the structure of XG. It is well known that the viscosity of polymeric solutions, both XG and HPAM, decrease with aging (time, temperature, pressure, etc.) due to transitional changes of the molecule and its subsequent polymeric degradation. In previous works [18], it was shown that the presence of nanoparticles increases the viscosity of XG polymeric solutions, increasing the hydrodynamic radius of the polymer, which is a sign of efficient polymer/nanoparticle interaction. These rheological results could be attributed to weak physical attractive forces between the components, such as hydrogen bonds. The surface of silica nanoparticles exhibits silanol groups (Si-OH) while XG chains contain hydroxyl groups (-OH), which makes dipole–dipole interactions possible [18,28].

In addition, it is reasonable to assume that a degradation of the polymers (XG and HPAM) occurs during the core flooding test due to the high pressures and high temperatures in the porous medium (accelerated aging). In this case, as shown in Figure 5, the nanoparticles limit the movement of the polymeric molecule, thus delaying its degradation. This implies maintaining the viscous properties of nanofluid for a longer time, and the resulting flow profile, could shut off high-permeability zones. Consequently, the viscous fingering effects are dampened, leading to a better volumetric sweep efficiency, so that oil is more effectively produced.

## 4. Conclusions

EOR methods based on polymer flooding are well established, but new challenges always emerge, which give impulse to the search for new solutions. Polymeric nanofluids of xanthan gum and silica nanoparticles represent a very attractive alternative to these techniques because they can, simultaneously, provide an increase in water viscosity and maintain this viscosity at aging conditions, both of which are beneficial to the efficiency of the process.

After evaluating the potential of nanofluids (XG + NP-SiO_2_) and comparing them with the most popular polymer flooding in the EOR industry (HPAM and XG), the present work concluded:There are no variations in the brine-mineral oil interfacial tension due to the presence of XG polymer or silica nanoparticles. Therefore, stable emulsions between nanofluids and oil may not be promoted.The evaluated nanofluids were superior to HPAM and XG polymer solutions, exhibiting better viscosity profile, rock saturation tests, and displacement efficiency. The polymeric solutions of HPAM and XG could have had an accelerated aging degradation during the core flooding test, given the temperature and pressure conditions. The nanoparticles were able to delay the aging of the XG polymer structure and lead to a better performance of the nanofluids in the flooding process.

## Figures and Tables

**Figure 1 nanomaterials-13-00925-f001:**
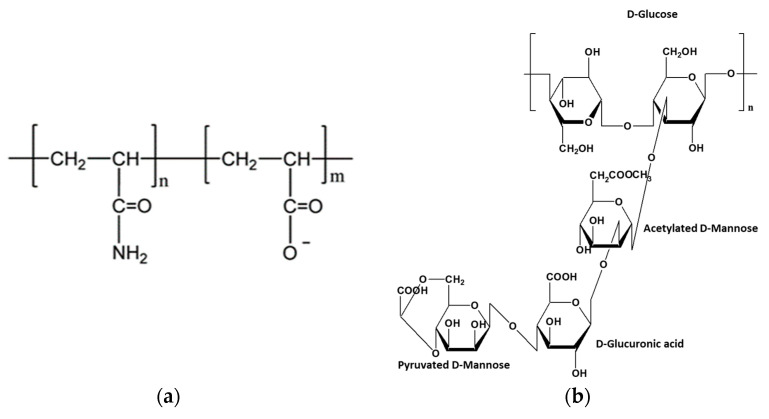
Molecular structure of (**a**) HPAM and (**b**) xanthan gum.

**Figure 2 nanomaterials-13-00925-f002:**
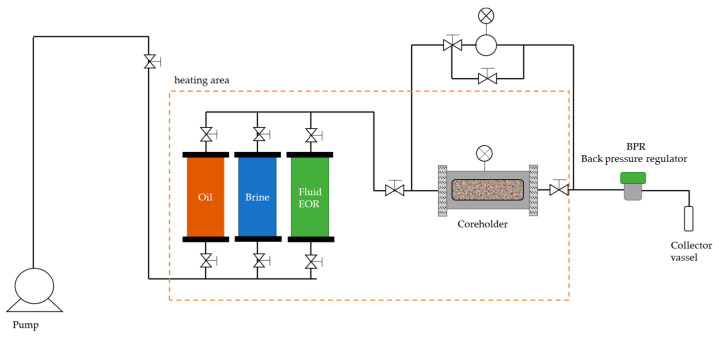
Core flooding experimental setup.

**Figure 3 nanomaterials-13-00925-f003:**
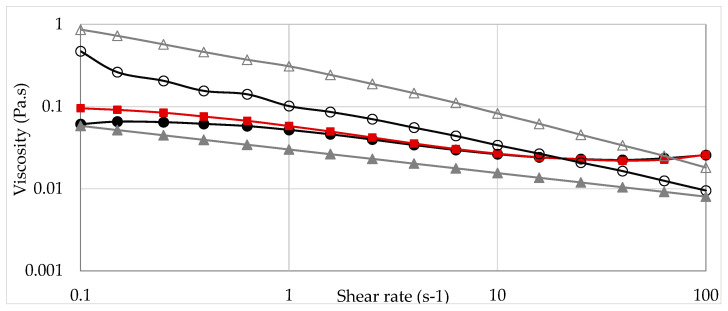
Viscosity of solutions with and without NaCl as a function of shear rate. HPAM solution in water (1000 ppm HPAM) (Δ); HPAM solution in brine (1000 ppm HPAM, 3 % NaCl) (▲); XG solution in water (1000 ppm XG) (○); XG solution in brine (1000 ppm XG, 3 % NaCl) (●); Nanofluid (1000 ppm XG, 300 ppm NP-SiO_2_, 3 % NaCl) (■).

**Figure 4 nanomaterials-13-00925-f004:**
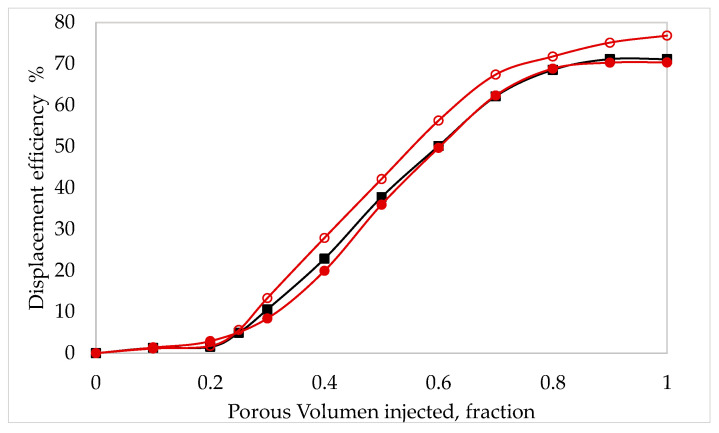
Displacement efficiency (%) as a function of porous volume injected. XG solution (1000 ppm XG, brine 3 % NaCl) (–●); HPAM solution (1000 ppm HPAM, brine 3 % NaCl) (–■); Nanofluid (1000 ppm XG, 300 ppm NP-SiO_2_, brine 3 %NaCl) (–○).

**Figure 5 nanomaterials-13-00925-f005:**
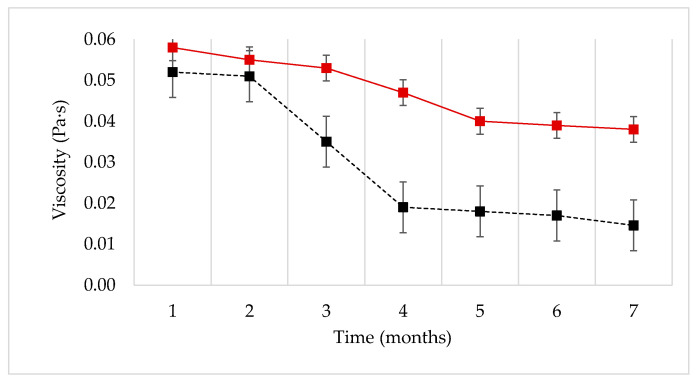
Viscosity as a function of time at a shear rate of 7.3 s^−1^. Storage and measurement at T=60 °C. Nanofluid (1000 ppm XG, 300 ppm NP-SiO_2_ and 3% NaCl) (–■). Polymer solution (1000 ppm XG and 3% NaCl) (-·-■) [18].

**Table 1 nanomaterials-13-00925-t001:** Interfacial tension.

Fluid	IFT with Paraffin (mN m^−1^) ^1^	σ (mN m^−1^) ^2^
Brine (3% NaCl)	24.2	0.91
Polymer solution	25.8	1.47
Nanofluid	24.7	1.06

^1^ Measurements performed at T = 60 °C. Nanofluid (1000 ppm XG and 300 ppm NP-SiO_2_) in 3% NaCl brine. Polymer solution (1000 ppm XG) in 3% NaCl brine. Paraffin: model oil. ^2^ σ standard deviation.

**Table 2 nanomaterials-13-00925-t002:** Core flood experiments.

Parameter	Core 1	Core 2	Core 3	Core 4
	Brine (3 % NaCl)	XG Solution	HPAM Solution	Nanofluid
Core conditions				
Core length (inches)	29.7	29.18	29.18	29.49
Pore volume (cm^3^)	80.51	82.2	89.5	78.84
Effective porosity (%)	22.5	23.39	25.46	22.20
Pore volume (cm^3^)	80.51	82.2	89.5	78.84
Saturations conditions				
Absolute permeability (D)	3.1	2.7	2.3	2.7
Swirr (%)	17.5	26.33	25.59	25.76
Residual oil saturation (%)	26.0	22.0	15.0	21.5
Flooding				
Initial oil saturation (%)	82.5	73.67	74.41	74.24
Displacement efficiency (%)	63.61	70.27	71.11	76.79
Incremental laboratory EOR (%)	-	6.6	7.5	13.18

## Data Availability

The data presented in this study are available on request from the corresponding author.

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
