# Peer review of "Silica Nanoparticles in Xanthan Gum Solutions: Oil Recovery Efficiency in Core Flooding Tests"

_nanomaterials, 2023, doi:10.3390/nano13050925_

Round 1

Reviewer 1 Report

The manuscript “Silica Nanoparticles in Xanthan Gum Solutions: Oil Recovery Efficiency in Core Flooding Tests” was studied in the Enhanced oil recovery (EOR) field, and core flooding experiments were conducted to further study the potential of nano fluid (XG+NP-SiO2) to improve oil recovery, and compared with traditional methods. I think this work has certain engineering value. This manuscript needs minor revision before considering published at Nanomaterials. Here are some questions to discuss with the author.

1.     The author has a lot of details in the writing of the article, which will make readers uncomfortable or confused. I hope the author will check the whole article and make corrections or explanations. For example, in the text in Figure 1, what does the “(xx)” after the two substances mean? And after the introduction of “Xanthan Gum (XG)”, the substance can be described by XG, without the full name. Some paragraphs in Introduction do not have full stops.

2.     In addition to the mistakes in details, the introduction is still very good, but I recommend that the author should pay attention to and quote some cutting-edge literature in related fields, such as the application of theoretical methods in EOR. This will enrich the article. For example, Chemical Engineering Science, 2022, 261: 117957. Chemical Engineering Science, 2020, 227: 115927.

3.     In the theoretical part, if a symbol represents a physical quantity, strictly speaking, italics should be used, such as pressure delta “” and permeability “K”. For others, the author should check himself.

4.     The new oil displacement method proposed by the author in the article has many advantages. What are the obstacles to industrial application? Are there any hidden dangers or shortcomings not mentioned by the author? Compared with the existing mainstream oil displacement methods, do they have some disadvantages?

5.     What effect does the use of silica nanoparticles with different specifications have on the results? Is there an optimal specification of silica nanoparticles?

6. Now that the author has explained that this method of oil displacement can improve oil recovery, can we add a short paragraph of analysis on industrial application in the paper. For example, how much more oil can be recovered by using new methods for an oilfield with specific specifications? How much economic benefit will this bring? How much cost is reduced at the same time? This can improve the engineering value of the article.

7. In Part 3.3, why begin by saying that the difference between XG and HPAM is because HAPM is less elastic and then HPAM is more elastic than XG. (Lines 319-322)

Reviewer 2 Report

The paper is well-written and the experiments appear to have been carefully carried out. One curious point is that while the authors describe the rheology of both XG and HPAM without and with nanoparticle (NP) and also the corefloods with XG, HPAM and (XG+NP), they don't mention anything about the coreflood with (HPAM+NP). In view of their observation that the viscosity increase due to NP addition is bigger for HPAM than XG, this makes the reader wonder why.  Some minor points are:

1. What is the avg size of NPs? Is their surface bare or coated (zeta potential?). If it's bare, are they stable in 3% NaCl?

2. What is the avg MW of XG and hydrodynamic radius?

3. For Fig. 5, what are the oxygen and Fe concentrations?

4. In Table 1, what does the last column represent?

5. In Table 2 and line 278, are the permeabilities in mD or D? For Ottawa sandpacks, they seem to be much too low.  
